# Overcoming Catastrophic Forgetting for Continual Learning via Model Adaptation

Wenpeng Hu[1,2,*], Zhou Lin[1,*] , Bing Liu[3,*], Chongyang Tao[2], Zhengwei Tao[2], Dongyan Zhao[2], Jinwen Ma[1], and Rui Yan[2,†]

[1]Department of Information Science, School of Mathematical Sciences, Peking University
[2]ICST, Peking University, Beijing, China
[3]Department of Computer Science, University of Illinois at Chicago
{*wenpeng.hu,jokerlin,chongyangtao,tttzw,zhaody,ruiyan*}*@pku.edu.cn*    *liub@uic.edu*
*jwma@math.pku.edu.cn*

## Abstract

Learning multiple tasks sequentially is important for the development of AI and lifelong learning systems. However, standard neural network architectures suffer from catastrophic forgetting which makes it difficult for them to learn a sequence of tasks. Several continual learning methods have been proposed to address the problem. In this paper, we propose a very different approach, called *Parameter Generation and Model Adaptation* (PGMA), to dealing with the problem. The proposed approach learns to build a model, called the *solver*, with two sets of parameters. The first set is shared by all tasks learned so far and the second set is dynamically generated to adapt the solver to suit each test example in order to classify it. Extensive experiments have been carried out to demonstrate the effectiveness of the proposed approach.

## 1 Introduction

It is well-known that neural networks (NNs) suffer from *catastrophic forgetting* (CF) (McCloskey & Cohen, 1989), which refers to the phenomenon that when learning a sequence of tasks, the learning of each new task may cause the NN to forget the models learned for the previous tasks. Without solving this problem, an NN is hard to adapt to *lifelong or continual learning*, which is important for AI.

**Problem Statement:** *Given a sequence of supervised learning tasks* $\mathbf{T} = (T_1, T_2, \ldots, T_N)$*, we want to learn them one by one in the given sequence such that the learning of each new task will not forget the models learned for the previous tasks.*

In recent years, many approaches (often called *continual learning*) have been proposed to lessen the effect of CF (Chen & Liu, 2018; Parisi et al., 2018), e.g., dynamically expandable network (DEN) (Yoon et al., 2018), learning without forgetting (LWF) (Li & Hoiem, 2016)), elastic weight consolidation (EWC) (Kirkpatrick et al., 2017), incremental moment matching (IMM) (Lee et al., 2017), gradient episodic memory (GEM) (Lopez-Paz et al., 2017), generative replay (GR) (Shin et al., 2017), etc. These existing studies except DEN and LWF focused on learning a model parameterized by a single joint set of parameters $\theta^*$ which is assumed to work well for all tasks. We call them *joint parameterization* (JP) methods. DEN and LWF require constantly increasing the number of parameters and thus can result in a huge and complex model.

JP methods, however, suffer from *accuracy deterioration*. Assume we have two tasks A and B that need to be learned sequentially. Let $\theta_A^*$ and $\theta_B^*$ be the optimal parameters for Task A and Task B respectively when each of them is learned individually. Let $\theta^*$ be the joint parameters learned by the existing approaches to perform Tasks A and B in sequence. Inevitably, $\theta^*$ is different from $\theta_A^*$ and/or $\theta_B^*$ and is highly likely to result in more errors for the two tasks than $\theta_A^*$ and $\theta_B^*$ individually.

---

*Equal contribution
†Corresponding author

In this paper, we propose a different approach, called *Parameter Generation and Model Adaptation* (PGMA), to dealing with CF and to significantly reducing accuracy deterioration. PGMA does not learn a joint parameter set $\theta^*$. Instead, it learns a parameter generator $f(\cdot)$ and a shared parameter set $\theta_0$. The key idea of PGMA is as follows: The overall classification network, called the *solver $S$*, has two disjoint subsets/parts of parameters. The first subset is $\theta_0$, which is shared by all tasks learned so far. The second subset is just a place holder $H$ that will be filled by parameters generated by $f(\cdot)$ for each test instance. That is, given a test instance, a set of parameters will be generated by the learned parameter generator $f(\cdot)$ to replace $H$. Solver $S$ then combines $\theta_0$ and the generated parameters for the test instance to classify it. Clearly, unlike existing JP approaches, we do not have a network with a set of fixed parameters for $S$. The idea is that $\theta_0$ contains the common features of all tasks, and the generated parameters for each test instance adapt $S$ for each test instance in order to classify it.

Since $f(\cdot)$ and the shared parameters $\theta_0$ will change during training for each new task, forgetting can occur for previous tasks. To deal with it, in training $f(\cdot)$ and $S$ for each task $T_i$, in addition to the training data of $T_i$, a small number of replayed samples will be generated by a data generation network for the previous tasks to ensure that the knowledge learned for previous tasks remain stable/unforgotten. Compared with the existing generative replay methods, our method does not need labels for the replayed data because we use the replayed data only to constrain the training of $S$ and $f(\cdot)$ rather than to treat them as surrogates of labeled training data from previous tasks. Further, since in the existing replay methods, the labels are produced by the current learned network itself, the labels can be noisy and biased, which may result in errors being accumulated and propagated to subsequent tasks. Our method does not have this problem.

Apart from reducing the effect of accuracy deterioration of the existing approaches, the proposed approach also has some other advantages. First, no parameter increase or network expansion is needed to learn new tasks. Second, no previous data needs to be stored to enable the system to remember the previously learned models or knowledge.

Experiments conducted using two image datasets (MNIST and CIFAR-10) and two text datasets (DBPedia ontology (Lehmann et al., 2015) and THUCNews (Li et al., 2006)) show that the proposed approach PGMA works well for different scenarios and different types of datasets, and outperforms the existing strong baselines markedly.

## 2 PROPOSED PGMA FRAMEWORK

Let the sequence of supervised learning tasks be $\mathbf{T} = (T_1, T_2, \ldots, T_N)$. Each task $T_i$ is represented by $T_i = \{x_{ij}, y_{ij} | j \in (1, \ldots, N_i)\}$, where $x_{ij}$ is the $j$-th example/sample of $T_i$, and $y_{ij}$ is its label. We use $(x^t, y^t)$ to denote a test instance/example. To simplify the notation, we will just use $(x_i, y_i)$ to denote a training example from task $T_i$, omitting the second subscript $j$. Note that in the paper, we use the terms *example*, *sample*, and *instance* interchangeably.

The proposed parameter generation and model adaptation (PGMA) architecture has three main components:

● **Solver $S$**: It is the main classification model. As mentioned in Section 1, the parameter set of $S$ consists of two subsets, $\theta_0$ that is shared by all tasks (and instances) and $H$, a parameter place holder, which will be replaced by the generated parameters set $\mathbf{p}_i$ (or $\mathbf{p}^t$) for each training (or testing) example $x_i$ (or $x^t$). $\mathbf{p}_i$ (or $\mathbf{p}^t$) basically serves to *adapt* the solver $S$ to classify the example in training or testing. This is the key idea of our approach. We adopt this parameter split as several studies (Yoon et al., 2018; Li & Hoiem, 2016; Kirkpatrick et al., 2017) have shown that only part of parameters of a neural network needs to be adjusted when learning a new task. See Section 2.2
● **Dynamic Parameter Generator (DPG)** $f(\cdot)$: It takes the embedding $\mathbf{z}_i$ (or $\mathbf{z}^t$) of each input training (or testing) example $x_i$ (or $x^t$) to generate the parameters $\mathbf{p}_i$ (or $\mathbf{p}^t$) for solver $S$. Note that we use $\mathbf{z}_i$ (or $\mathbf{z}^t$) rather than the raw data $x_i$ (or $x^t$) because the raw data's dimension can be very high. The embedding $\mathbf{z}_i$ (or $\mathbf{z}^t$) as a low dimensional dense representation reduces the mapping space for DPG and thus reduces the difficulty in its parameter generation. See Section 2.2.
● **Data Generator (DG)**: It has two functions. The main function is to generate a set of replayed data or samples $\{x'_m\}_{m=1}^M$ using its decoder $DG_D$ for previous tasks to deal with catastrophic forgetting. The other function is to generate the embedding $\mathbf{z}_i$ (or $\mathbf{z}^t$) of each input training (or testing) example $x_i$ (or $x^t$) using its encoder $DG_E$. See Section 2.3.

## 2.1 OVERALL APPROACH OF PGMA

We work backward by describing testing first before training. Given a test instance $x^t$, $DG_E$ first generates its embedding $\mathbf{z}^t$, which is fed to $f(\cdot)$ to generate a set of parameters $\mathbf{p}^t$. Solver $S$ then takes $x^t$ as input and uses the trained/learned shared parameters $\theta_0$ and $\mathbf{p}^t$ to classify $x^t$. $\theta_0$ contains the common features of all tasks learned so far. $\mathbf{p}^t$ simply adapts $S$ for $x^t$ in order to classify $x^t$.

For training, the pipeline of the proposed PGMA framework is shown in Figure 1. Given a new task $T_i$ with its data $(x_i, y_i)$, solver $S$ and DPG $f(\cdot)$ are jointly trained to learn $T_i$ and also not to forget the previously learned tasks. In each iteration, a set of parameters $\mathbf{p}_i$ is generated by the current DPG $f(\mathbf{z}_i, \mu)$ for each training instance $x_i$ ($\mathbf{z}_i$ being its embedding), where $\mu$ is the set of parameters of the DPG network, which is trained.

In training the solver $S$ and DPG for the new task $T_i$, both $f(\cdot)$ and the shared parameters $\theta_0$ will change, which can cause forgetting in DPG for previous tasks. To keep DPG remembering the acquired knowledge for previous tasks, we minimize the variation of certain layers' output caused by the changes of $\theta_0$ and $f(\cdot)$ using the set of replayed samples $\{x'_m\}_{m=1}^M$ generated by DG.

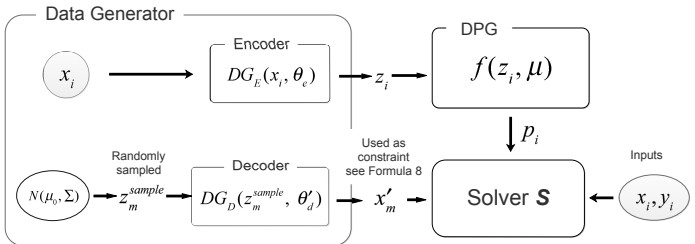

Figure 1: Sequential training in the PGMA framework.

PGMA has two training components: (1) DPG and solver $S$ (**DPG&S** for short), i.e., DPG is optimized together with solver $S$. (2) **DG**. The two components have their respective objective functions (see Sections 2.2 and 2.3), and are trained alternately. This is because DPG takes input $\mathbf{z}_i$ (which is produced by DG) to generate parameters, and alternating training ensures consistent convergence rates of DG and DPG.

## 2.2 DYNAMIC PARAMETER GENERATOR (DPG) AND SOLVER $S$ IMPLEMENTATION

Several neural networks can be used to implement DPG for parameter generation, e.g., convolutional neural network (CNN), recurrent neural network (RNN) and multilayer perceptron (MLP). The objective of this paper is not to explore all possible implementations. We found that a straightforward implementation using MLP can already achieve good results. Formally, DPG can be written as:

$$\mathbf{p}_i = f(\mathbf{z}_i, \mu) = \sigma(\mathbf{w}_D \mathbf{z}_i + \mathbf{b}) \tag{1}$$

where $\sigma$ is the activation function, and $\mathbf{w}_D$ and $\mathbf{b}$ are the parameters of DPG and denoted by $\mu$.

For the implementation of solver $S$, several deep learning networks can be used as well. Again, a MLP is employed in this work. Each layer of the MLP is a perceptron and can be formalized by $output = \sigma(\mathbf{w}\mathbf{x}_{input})$, where $\mathbf{x}_{input}$ denotes the input of the particular perceptron. We also call a perceptron a *basic unit* [1]. In general, for each basic unit $k$ of the solver $S$, we can have a shared portion of the parameters $\theta_{0,k}$ and a generated portion of the parameters $\mathbf{p}_{i,k}$, i.e.,

$$\theta_i^* = \text{combine}(\theta_0, \mathbf{p}_i) = \{[\mathbf{w}_{i,k}^*]\}_{k=1}^K = \{[\theta_{0,k}; \mathbf{p}_{i,k}]\}_{k=1}^K \tag{2}$$

where combine(.) is concatenation and $K$ is the number of basic units that the solver has. In our case, $K$ is the number of hidden layers (basic units) of the solver MLP.

It is important to note that the above is the most general case. In practice, there is no need to adapt all the basic units in the solver, but only a subset of them. In our experiments (Section 3), we will see that adapting only the final layer of the solver MLP can already achieve good results.

---

[1]This is because most of the existing networks can be regarded as the composition of multiple basic units, even for those with complex structures, e.g., seq2seq (Sutskever et al., 2014), R-Net (Wang et al., 2017), Resnet (He et al., 2016) and Densenet (Huang et al., 2017). Further, complex structures give us more freedom to decide which part of the parameters should be generated.

However, one may still ask whether the proposed combination method has sufficient capacity to adjust the solver $S$ in general because only part of the parameters are generated. Actually, $\mathbf{p}_{ik}$ can affect all dimensions of the output of its corresponding basic unit $k$:

$$\mathbf{w}_{i,k}^* \mathbf{x}_{input} = [\theta_{0,k}; \mathbf{p}_{i,k}] \mathbf{x}_{input} = \theta_{0,k} \mathbf{x}_{input}^1 + \mathbf{p}_{i,k} \mathbf{x}_{input}^2 \tag{3}$$

where $\mathbf{x}_{input}^1$ and $\mathbf{x}_{input}^2$ are the block vectors of $\mathbf{x}_{input}$. $\mathbf{p}_{i,k} \mathbf{x}_{input}^2$ can be regarded as the bias and can adapt the output vector to any point in the vector space.

To train DPG&S, we use cross entropy loss ($\mathcal{L}_{ce}$). The objective function of the solver $S$ including DPG $f(\mathbf{z}_i, \mu)$ and $\theta_0$ for learning each new classification task $T_i$ is defined as:

$$\underset{\mu, \theta_0}{\text{minimize}} \quad \mathcal{L}_{ce}(S(x_i, \theta_i^*), y_i)$$

$$\text{s.t.} \quad \sum_{m=1}^{M} ||\mathcal{R}(x_m', \theta_i^*) - \mathcal{R}(x_m', \theta_{i-1}^*)|| < \epsilon_r \tag{4}$$

where $\mathcal{L}_{ce}$ is the cross entropy loss, $\mathcal{R}(\cdot)$ denotes the output of all basic units (which can be any layer or a set of layers[2]) in solver $S$, which we will introduce later, and $\theta_i^*$ is the whole set of parameters of $S$ ($\theta_i^* = \text{combine}(\mathbf{p_i}, \theta_0)$) representing the adaptation of $S$. We can see that the generated replayed samples $x_m'$ are used as constraints to alleviate DPG&S's forgetting. Specially, we extend the knowledge distillation loss (Hinton et al., 2015) to the general situation. That is, to keep the past learned knowledge, the output of the basic units in the solver should not change much when learning a new task with the help of the generated data. If we do not consider the activation function, the constraints in Eq. 4 can also be written as:

$$\min \sum_{m=1}^{M} \sum_{k=1}^{K} ||\mathbf{w}_{i,k}^* \mathbf{x}_{m,k}' - \mathbf{w}_{i-1,k}^* \mathbf{x}_{m,k}'|| \tag{5}$$

where $K$ again denotes the number of basic units and $M$ denotes the number of replayed samples. The basic unit with a smaller $k$ is at the relative lower layer of the solver network.[3] $\mathbf{x}_{m,k}'$ is the input of the $k^{th}$ basic unit and is calculated through forward propagation, except $\mathbf{x}_{m,1}' = DG_D(\mathbf{z}_m^{sample}, \theta_d')$ which is the initial replayed sample $x_m'$ generated by DG (before optimizing the current task $T_i$). $\mathbf{w}_{i,k}^*$ is the combined parameter introduced in Section 2.2.

Two questions may be asked about Eqs. 4 and 5. (1) Since the replayed data $x'$ (denoting all $x_m'$) is used in Eqs. 4 and 5, does it play the same role as the original data $x$ from previous tasks? (2) What is the impact of the constraints on learning the new task? We answer the two questions now.

*Question* 1: The answer is positive. Since the purpose of the constraints is to maintain the learned effect of the old data, clearly, using the original old data $x$ in the constraints is better. However, we proved the positive answer to question 1, which is given in Appendix.

*Question* 2: In calculating Eq. 5, if we stack all $\{\mathbf{x}_{m,k}'\}_{m=1}^{M}$ (column vectors) into a matrix $\mathbf{X}_k'$, Eq. 5 can be regarded as constraining $\mathbf{w}_{i,k}^* \mathbf{X}_k'$ to remain unchanged. If $\mathbf{X}_k'$ is a row low rank matrix, $\mathbf{w}_{i,k}^*$ can be trained to fit the new tasks. Otherwise, especially if $\mathbf{X}_k'$ is a row full rank matrix, $\mathbf{w}_{i,k}^*$ cannot be trained and therefore cannot learn new tasks. However, benefiting from dynamic parameter generation, our approach will not suffer from this problem. That is because $\mathbf{w}_{i,k}^*$ is specially generated through DPG to perform well with input $\mathbf{X}_k'$. And the new parameters will be generated to fit new tasks while $\mathbf{w}_{i,k}^*$ is able to remain unchanged.

## 2.3    DATA GENERATOR (DG)

As indicated earlier, DG has two functions. First, it compresses the original input data $x_i$ to $\mathbf{z}_i$ using its encoder to reduce the number of dimensions of $x_i$ and consequently reduces the mapping space of DPG to make the generation of the parameters $\mathbf{p}_i$ easier. The compression is formulated by:

$$\mathbf{z}_i = DG_E(x_i, \theta_e) \tag{6}$$

---

[2]Note that in this case, the parameters in $\mathcal{R}$ is a subset of parameters ($\theta_i^*$) for solver $S$. To simplify the expression and without causing confusion, we don't introduce new symbols and let the symbols unchanged.

[3]Note that constraining only the units in the last layer can already achieve good results.

where $DG_E$ is the encoder of DG with parameters $\theta_e$.

Second, it generates the replayed data for previous tasks to deal with forgetting in solver $S$ and DPG. Note that DG differs from data generators in the existing generative replay (GR) methods (Shin et al., 2017) as GR needs to generate both the replay data $x'_m$ and their labels $y'_m$ (using the solver learned so far), which can be noisy. DG only generates the data $x'_m$ but not the labels (which are not needed by our approach), and then the labeling errors won't affect our model PGMA, but will hurt GR.

Each replayed sample $x'_m$ is generated by the decoder of DG, called $DG_D$:

$$x'_m = DG_D(\mathbf{z}^{sample}_m, \theta'_d) \tag{7}$$

where $\mathbf{z}^{sample}_m$ is the $m^{th}$ sample sampled from the multivariate normal distribution.[4] $\theta'_d$ is the set of parameters of $DG_D$ before optimizing the current task $T_i$.

DG can be implemented with an auto-encoder, e.g., VAE-like (Variational Auto-Encoder (Kingma & Welling, 2013)) and WAE-like (Wasserstein Auto-Encoder) auto-encoders. We use WAE in DG since it can let different examples get a chance to stay far away from each other, which promotes better reconstruction (Tolstikhin et al., 2017).[5] Note that DG also suffers from forgetting, which is dealt with like DPG and also using specially designed losses (see below).

To train DG, we use *mean square error* as the reconstruction loss to enable its replay ability of the past data, and add to it the penalized form of the Wasserstein distance between the distribution of $\mathbf{z}_i$ and multivariate normal distribution to help generate data (Tolstikhin et al., 2017) (together denoted as $\mathcal{L}_{wae}$).

Note also since we only have one data generator DG, it has the forgetting problem caused by incremental training of new tasks too. To avoid forgetting in DG, we investigated using loss functions to constrain DG to overcome its forgetting as shown in Figure 2. We describe this approach here:

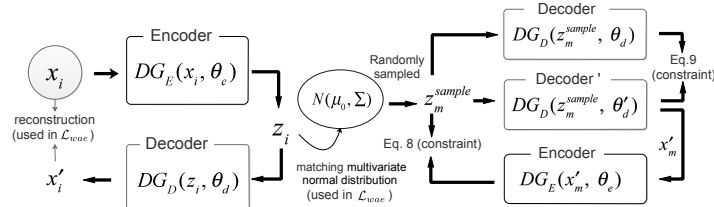

Figure 2: Data Generator training pipeline demonstration.

$$\min \sum_{m=1}^{M} ||\mathbf{z}^{sample}_m - DG_E(x'_m, \theta_e)|| \tag{8}$$

$$\min \sum_{m=1}^{M} ||DG_D(\mathbf{z}^{sample}_m, \theta_d) - x'_m|| \tag{9}$$

where $x'_m$ is the replayed data (see Eq. 7), $\theta_e$ and $\theta_d$ are the encoder's and decoder's parameters of DG in the process of learning the new task $T_i$, respectively, and $\mathbf{z}^{sample}_m$ and $M$ have the same meanings as they are in DPG&S. Eq. 8 constrains the consistency of DG's decoder and encoder over the randomly sampled $\mathbf{z}^{sample}_m$. Eq. 9 ensures that DG's decoder can still remember the old data. Using Eqs. 8 and 9, we can maintain DG's ability to reflect the old data. Overall, the final objective function for DG is composed by $\mathcal{L}_{wae}$, Eq. 8 and 9.

## 2.4 THE COMPLETE TRAINING PROCEDURE

Finally, we summarize the whole training procedure of PGMA in Algorithm 1, which is self-explanatory. "//" is followed by comments. The test procedure is straightforward, see Section 2.1.

---

[4]In training, we force $\mathbf{z}_i$ to satisfy the multivariate normal distribution. We can then sample many $\mathbf{z}^{sample}_m$ from the multivariate normal distribution.

[5]Our approach is not limited to using WAE. Other auto-encoders may also be applied.

---

**Algorithm 1** PGMA (Parameter Generation and Model Adaptation) training

**Input:** $\mathbf{T} = \{T_i\}_{i=1}^N$, where $T_i = \{x_{ij}, y_{ij}\}$     15: // Learning the subsequent tasks
**Initial:** Randomly initialize $f(\cdot)$, $DG$, $S$;        **for all** $i = 2$; $i \leq N$; $i++$ **do**
// Learning the first task                   Generate replayed samples $x'_m$ by $DG$;
**for all** $n = 0, \dots$, until convergence **do**       **for all** $n = 0, \dots$, until convergence **do**
5:    Sample a mini-batch from $T_1$;               Sample a mini-batch from $T_i$;
// Training DG          20:      Minimize $\mathcal{L}_{wae}$, Eqs. 8 & 9 and then update
Minimize $\mathcal{L}_{wae}$ and then update $DG$;            $DG$;     // Section 2.3
// Training DPG and $S$. $\mathbf{p}_n$ and $S_n$ below        Compute $\mathbf{p}_n$ using Eqs. 1 & 6;
// denote a batch of generated parameters         Construct $S_n$ using $\mathbf{p}_n$ and $\theta_0$;
10:    // and adapted solvers, respectively           Minimize $\mathcal{L}_{ce}$, Eq. 5, and then update $f(\cdot)$
Compute $\mathbf{p}_n$ using Eqs. 1 & 6;    // Section 2.2         and $S$;    // Section 2.2
Construct $S_n$ using $\mathbf{p}_n$ and $\theta_0$;    // Section 2.2     **end for**
Minimize $\mathcal{L}_{ce}$ then update $f(\cdot)$ and $S$;     25: **end for**
**end for**

---

# 3 EXPERIMENTS

We now evaluate the proposed approach PGMA[6] and compare it with state-of-the-art baselines using two image datasets and two text datasets.

**Datasets**

• **Two image datasets**: (1) MNIST: this dataset consists of 70,000 images of handwritten digits from 0 to 9. We use 60,000/3000/7000 images for training/validation/testing respectively. (2) CIFAR-10: this dataset consists of 60,000 32x32 color images of 10 classes, with 6000 images per class. There are 50,000/3000/7000 images for training/validation/testing respectively.
• **Two text datasets**: (1) DBPedia ontology: this is a crowd-sourced dataset (Lehmann et al., 2015) with 560,000 training samples and 70,000 test samples. Out of the 70,000 test samples, we use 10,000 for validation and 60,000 for test. (2) THUCNews: this dataset consists of 65,000 sentences of 10 classes (Li et al., 2006). We randomly select 50,000/5000/10,000 sentences for training/validation/testing respectively.

**Experiment Settings**

**Data Preparation:** To simulate sequential learning, we adopt the same two data processing methods as in (Lee et al., 2017), named *disjoint* and *shuffled*.
• Disjoint: This method divides each dataset into several subsets of classes. Each subset is a task. For example, we divide the MNIST dataset into two tasks (or subsets of classes). The first task consists of digits (classes) $\{0, 1, 2, 3, 4\}$ and the second task consists of the remaining digits (classes) $\{5, 6, 7, 8, 9\}$. The systems learn the two subsets as two tasks in a sequential fashion and regard them together as 10-class classification. In order to consider more tasks in testing, for MNIST, CIFAR-10, and THUCNews, which all have 10 classes, we created two experiment settings of 2 tasks (5 classes per task) and 5 tasks (2 classes per task). For DBPedia, which has 14 classes, we created three experiment settings, 2 tasks (7 classes per task), 3 tasks (5, 5, and 4 classes for the three tasks respectively), and 5 tasks (3, 3, 3, 3, and 2 classes for the 5 tasks respectively).
• Shuffled: This method shuffles the input pixels of an image with a fixed random permutation. Two experiment settings were created: 3 tasks and 5 tasks. In both cases, the dataset for the first task is the original dataset. The datasets for the rest of the tasks are constructed through shuffling. Since shuffling of words in a sentence will change the sentence meaning and results in confusion, thus this experiment is not done on text datasets.

**Baselines:** We use three state-of-the-art baselines that are representative of the current approaches: 1) **EWC** (elastic weight consolidation) (Kirkpatrick et al., 2017); 2) **IMM** (incremental moment matching) (Lee et al., 2017); 3) **GR** (generative replay) (Shin et al., 2017). We use the open source code released by the authors or the third party for comparison. We use Adam algorithm to update the parameters and also use Adam as a baseline to show how serious the forgetting problem is.

---

[6]https://github.com/morning-dews/PGMA_tensorflow

**Training Details:** For fair comparison, our proposed approach uses the same solver (or classifier) as the baselines. That is, a multilayer perceptron is adopted as the solver/classifier (as the baselines all use this method), which is a 3-layer network (i.e., two basic units with each hidden layer as a unit) followed by a softmax layer. For our approach, the total number of parameters in the solver includes both the generated parameters $\mathbf{p}$ and the shared parameters $\theta_0$. Due to the differences among different datasets, we adopt different settings for them, see Table 5 in *Appendix* for details. All baselines and our approach use the same setting for the same dataset. We use a 3-layer perceptron (with 2 hidden layers) network (we also call it *T-net*) for DPG and set the size of each hidden layer to 1000. Each T-net can generate 100 parameters at a time. We can parallel several T-nets in the DPG to generate more parameters when needed. The network parameters are updated using the Adam algorithm with a learning rate of 0.001.

### Results and Analysis

Figure 3 shows the test accuracy plots per 40 training steps of each method for each task as the tasks are sequentially learned. IMM is not shown here because IMM works by combining well trained models of individual tasks to form one joint model for all tasks. Thus, we cannot draw an accuracy curve with increased training steps like others. However, its final results and those of the other systems are given in Tables 1 and 2. Shuffled CIFAR-10 is also not included in the figure because our experiments show that it cannot be learned by CNN. It is also not used in any existing paper. Note that due to space limitations, Figure 3 only plots the results of those settings with only 2 and 3 tasks.

From Figure 3, we can make the following observations:

(1). The proposed PGMA method consistently outperforms the baselines in overcoming forgetting, by a big margin in most cases.

(2). EWC does not perform well for the disjoint setting. GR is better but still poorer than our method. Adam's results show that forgetting is very serious for all datasets and settings.

**Time efficiency:** Due to DPG, our method uses slightly more time than baselines, under 25% for training and under 28% for testing compared to GR, which is efficient. This is a small price to pay for the major gain in accuracy in dealing with catastrophic forgetting.

**Memory usage:** There is no need for our method to save data, which results in a large memory saving. For more details about time efficiency and memory usage, please see *Appendix* (Section C).

Table 1: Average accuracy over all tasks in a sequence after the tasks have all been learned.

| Model | shuffled MNIST (3 tasks) | disjoint MNIST (2 tasks) | disjoint CIFAR-10 (2 tasks) | THUCNews (2 tasks) | DBPedia (2 tasks) | DBPedia (3 tasks) |
|---|---|---|---|---|---|---|
| Adam | 91.46 | 48.64 | 41.99 | 46.78 | 47.66 | 41.10 |
| EWC | 96.70 | 48.96 | 37.75 | 45.02 | 53.95 | 35.89 |
| GR | 97.57 | 89.96 | 65.11 | 81.55 | 88.41 | 82.14 |
| IMM | 97.92 | 94.12 | 62.98 | 80.32 | 92.65 | 85.31 |
| Our PGMA | **98.14** | **96.53** | **69.51** | **85.12** | **94.70** | **88.06** |

From Table 1, we can see that our PGMA method is consistently superior to EWC, GR, and IMM on different datasets with 2 or 3 tasks (5 tasks results are given in Table 2). We believe that is because our PGMA model can reduce the effect of the accuracy deterioration problem discussed in Section 1. Adam's results show that forgetting is very serious as it does not deal with the forgetting problem. EWC's performance is poor for the disjoint case (a more realistic setting in practice), which is also reported by other researchers (Lee et al., 2017). *Appendix* has the results for only the last task.

Table 2: Average accuracy over *5 tasks* in a sequence after the tasks have all been learned.

| Model | shuffled MNIST | disjoint MNIST | DBPedia | CIFAR-10 | THUCNews |
|---|---|---|---|---|---|
| GR | 94.54 | 75.47 | 63.71 | 31.09 | 47.35 |
| IMM | 96.09 | 67.25 | 64.04 | 32.36 | 46.61 |
| Our PGMA | **96.77** | **81.70** | **69.68** | **40.47** | **52.93** |

Table 2 shows the results for 5 tasks. EWC and Adam are not included as they performed poorly. Again, we observe that our PGMA method is markedly better than the baselines.

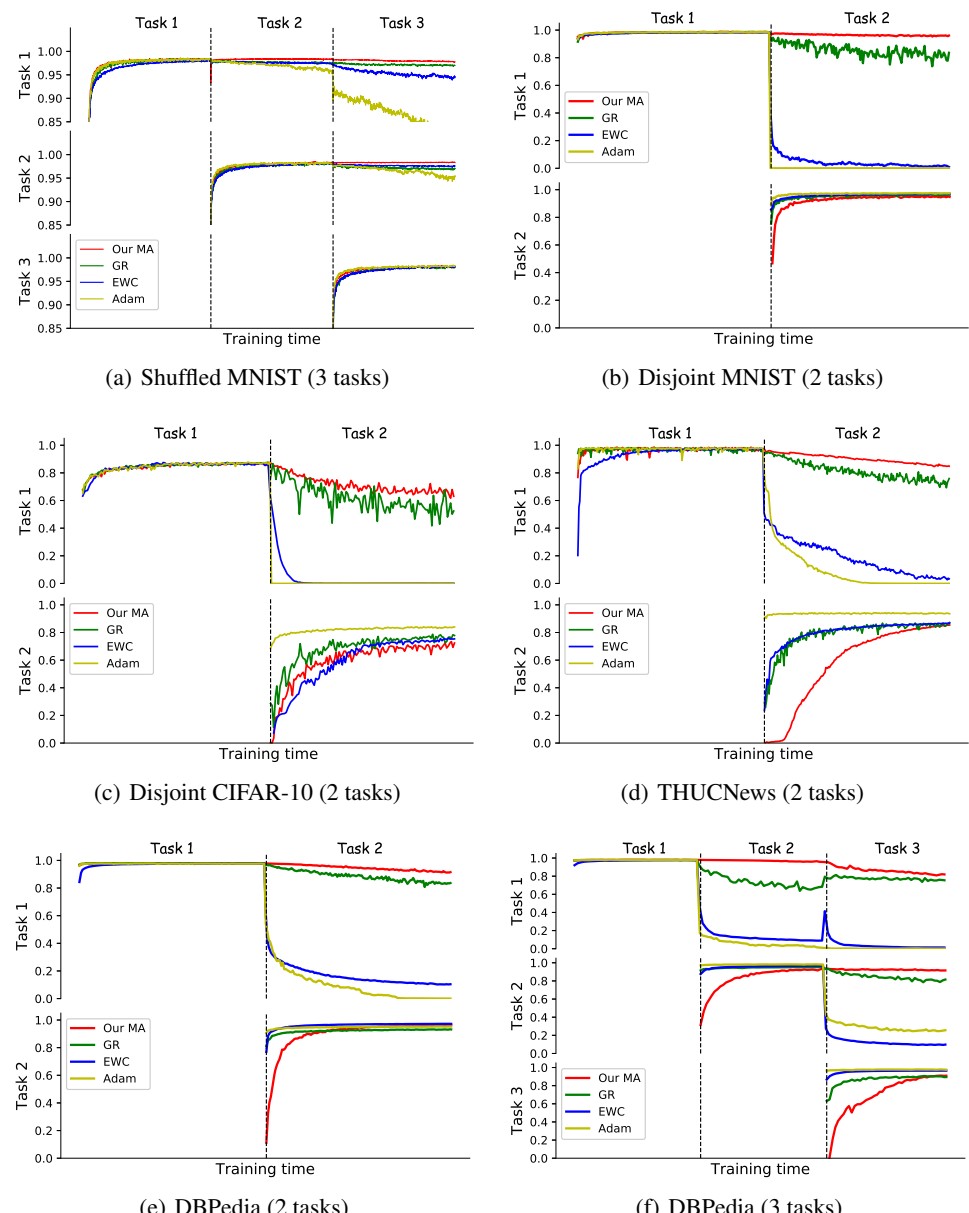

Figure 3: Accuracy curves - we test the system's accuracy per 40 training steps on the test set of each method. Y-axis shows the accuracy of different tasks as the training time (steps) increases. Note that Shuffled CIFAR-10 is not included as CNN cannot learn it directly.

**Ablation Study**

Here we study how the system behaves with less and less parameters in $\theta_0$ or more and more parameters replaced by the parameters generated by DPG. We selected the disjoint MNIST 2 tasks setting to conduct the experiment as it is more useful than the shuffled setting and also more difficult.

Table 3: Average accuracy with different % of parameters in solver's **last layer** (one basic unit) being replaced by the parameters generated by DPG. Each score followed by a 95% confidence interval.

| Model | GR | Our PGMA | | | | | |
|---|---|---|---|---|---|---|---|
| | | 0% | 20% | 40% | 60% | 80% | 100% |
| Accuracy | 88.71 (±2.64) | 90.96 (±0.69) | 92.44(±1.31) | 93.96(±0.63) | 94.51(±0.87) | 96.07(±0.62) | 96.05(±0.64) |

Table 3 shows the accuracy results (averaged in the same way as in Table 1) when a portion of the parameters in only the **last layer** of the solver is replaced by the parameters generated by DPG. We observe that the accuracy improves with increased percentages of parameters being replaced.

The best accuracy is obtained when 80% of the parameters in the last layer are replaced through DPG, and the accuracy won't further improve with more replaced parameters. This observation indicates that replacing a part of parameters in solver to adapt new input tasks is sufficient. The same conclusion can also be made by replacing the parameters in the first hidden layer of the solver (which has 2 hidden layers). We fix the replacing percentage of the last layer to 20%, and then increase the replacing percentages of the first layer. The best accuracy reaches 94.31%($\pm$0.84%) when replacing 40% parameters of the first layer, which gains only 1.87% in accuracy compared with no replacement. This result indicates that it suffices to replace the parameters in the last layer.

Table 4 shows the contribution of different components. We can see a significant drop if DPG is removed. The second row gives the performance of our model when only DG and constraints are used. The third row shows the result without using constraints (DG+label(like GR)) but replacing them with the replay method. We can see that constraints work better than predicting labels.

Table 4: Empirical evaluation of different components

| Components | Acc (%) |
|---|---|
| DPG + DG + Constraints | 96.07 $\pm$ 0.62 |
| DG + Constraints | 90.96 $\pm$ 0.69 |
| DG + label (like GR) | 88.21 $\pm$ 2.81 |

## 4  RELATED WORK

Many approaches have been proposed to deal with catastrophic forgetting (CF), which is one of the challenging problems of neural networks for lifelong learning (Chen & Liu, 2018). EWC (Kirkpatrick et al., 2017) quantifies the importance of weights to previous tasks, and selectively alters the learning rates of weights. Following EWC, Zenke et al. (2017) measured the synapse consolidation strength in an online fashion and used it as regularization. Learning without forgetting (LWF) (Li & Hoiem, 2016) feeds the old network with new training data in new tasks and regards the output as "pseudo-labels". In Incremental Moment Matching (IMM) (Lee et al., 2017), each trained network on one task is preserved and all networks are merged into one at the end of the sequence of tasks.

Above approaches focus on adjusting the network weights. Another main approach is to add some data of past tasks to the new task training to prevent forgetting the past. Gradient Episodic Memory (GEM) (Lopez-Paz et al., 2017) stores a subset of training data for every finished task, and limits the loss function on these so-called "memories". Instead of keeping some real data from previous tasks, Generative Replay (GR) (Shin et al., 2017) keeps data generators for previous tasks and learns using a mix of real data of the new tasks and replayed data of previous tasks. Seff et al. (2017) proposed to solve continual generative modeling by combining the ideas of data generation and EWC.

Other existing approaches include iCaRL (Rebuffi et al., 2017), Pathnet (Fernando et al., 2017), memory aware synapses (Aljundi et al., 2017), phantom sampling (Venkatesan et al., 2017), active long term memory networks (Furlanello et al., 2016), conceptor-aided backprop (He & Jaeger, 2018), gating networks (Masse et al., 2018; Serrà et al., 2018), dynamically expandable networks (DEN) (Yoon et al., 2018), progress & compress (Schwarz et al., 2018) (active column is distilled into the knowledge base, taking care to protect any previously acquired skills), and incremental regularized least squares (Camoriano et al., 2017). Most of those works suffer from accuracy deterioration as discussed in Section 1 while some dynamically expanding the network size (e.g., DEN and LWF), which results in a huge and complex model. Our method is different from these existing approaches.

Several existing works generate parameters, but they are for different purposes. Denil et al. (2013) trained several different architectures by learning only a small number of weights and predicting the rest. Ha et al. (2016) used a small network to generate the weights for a larger network. Brock et al. (2017) learned an auxiliary HyperNet to generate weights for the main model. Our method uses the generated parameters for model adaptation for continual learning.

## 5  CONCLUSION

This paper proposed a novel approach PGMA to dealing with catastrophic forgetting. The approach learns to build a model with two sets of parameters. The first set is shared by all tasks learned so far and the second set is dynamically generated to adapt the model (solver) to suit each individual test example. As we discussed in related work, this is different from all existing methods. Experimental results showed that the proposed approach outperformed the existing baseline methods markedly.

ACKNOWLEDGMENTS

We thank Zhangming Chan and Shen Gao for helping check the code. This work was supported by the National Key Research and Development Program of China (No. 2017YFC0804001). Bing Liu's work was partially supported by National Science Foundation (NSF) under grant no. IIS 1838770, by a research contract with Huawei Technologies Co. Ltd., and by a research gift from Tencent Holdings Limited. Rui Yan's work was supported by the National Science Foundation of China (NSFC No. 61672058; NSFC No. 61876196), CCF-Tencent Open Research Fund and Microsoft Research Asia (MSRA) Collaborative Research Program.

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

APPENDIX

## A  PROOF FOR QUESTION 1

*Proof:* Our objective is to minimize $||\mathcal{R}(x, \theta_i^*) - \mathcal{R}(x, \theta_{i-1}^*)||$. Since $\mathcal{R}(\cdot)$ is differentiable and its derivative function is bounded (as it uses 1-Lipschitz activation function, e.g., RELU or tanh), $R(\cdot)$ thus satisfies the Lipschitz condition. After some manipulation, we obtain:

$$
\begin{aligned}
&||\mathcal{R}(x, \theta_i^*) - \mathcal{R}(x, \theta_{i-1}^*)|| \\
=\ &||\mathcal{R}(x, \theta_i^*) - \mathcal{R}(x', \theta_i^*) + \mathcal{R}(x', \theta_i^*) - \mathcal{R}(x', \theta_{i-1}^*) + \mathcal{R}(x', \theta_{i-1}^*) - \mathcal{R}(x, \theta_{i-1}^*)|| \\
\leq\ &||\mathcal{R}(x, \theta_i^*) - \mathcal{R}(x', \theta_i^*)|| + ||\mathcal{R}(x', \theta_i^*) - \mathcal{R}(x', \theta_{i-1}^*)|| + ||\mathcal{R}(x', \theta_{i-1}^*) - \mathcal{R}(x, \theta_{i-1}^*)|| \\
\leq\ &\mathcal{L}_1||(x - x')|| + ||\mathcal{R}(x', \theta_i^*) - \mathcal{R}(x', \theta_{i-1}^*)|| + \mathcal{L}_2||(x - x')|| \\
=\ &\underbrace{||\mathcal{R}(x', \theta_i^*) - \mathcal{R}(x', \theta_{i-1}^*)||}_{\text{minimize replayed samples}} + (\mathcal{L}_1 + \mathcal{L}_2)||(x - x')||
\end{aligned}
\tag{10}
$$

where the last inference is based on Lipschitz continuity, and $\mathcal{L}_1$ and $\mathcal{L}_2$ are Lipschitz constants. We can see that if the solver has a small Lipschitz constant or the DG's reconstruction error is small, minimizing Eq. 4 is consistent with constraining using the original data.

## B  PARAMETER SETTINGS

Table 5: Parameter settings for different datasets

| |
|---|
| **MNIST**: We set the basic unit (hidden layer) size of the 3-layer classifier to 600 and the dropout rate to 0.3. A 3-layer convolutional network (CNN), with 2 * 2 convolutions and 64, 128, 256 filters for each layer, is used as DG's encoder. A deconvolutional network with the symmetric setting as the encoder is used as the decoder in DG. For our model, we use the parameters generated by DPG to replace 33% of the parameters in the solver's last hidden layer, and 10% of the first hidden layer. |
| **CIFAR-10**: We set the basic unit (hidden layer) size of the 3-layer classifier to 1000 and the dropout rate to 0.5. For our model, we use the parameters generated by DPG to replace 25% parameters of the solver's last hidden layer, and 10% of the first hidden layer. Since each image in this dataset contains a complex background and is thus more difficult to classify, we add a 3-layer CNN below the 3-layer MLP classifier to improve the performance. 3-layer CNN has the same setting as the encoder of DG. Due to the complexity of the images in CIFAR-10, the replayed images are usually noisy. Text datasets usually cannot be exactly replayed. To alleviate this problem, we propose to replay the features (e.g. using an additional CNN to extract features) of input data which performs well in the experiment. In that case, the 3-layer CNN added to the classifier plays an important role in feature extraction and it needs to be fixed; if not, the system won't get a stable input and thus will cause forgetting. Note that the feature extractor can be pre-trained using a large dataset (e.g. ImageNet for images and wikipedia for text.) and thus can provide sufficient features. |
| **Text Datasets**: We set the basic unit (hidden layer) size of the 2-layer classifier to 1000 and dropout rate to 0.5. For our model, we use the parameters generated by DPG to replace 25% parameters of the solver's last hidden layer, and 10% of the first hidden layer. To get better results on text, we use pre-trained embeddings (Pennington et al., 2014; Li et al., 2018) for all experiments. |

## C  MORE EXPERIMENTAL RESULTS

**Memory**

• The memory used by our model is around 151.49MB, by IMM is around 38.96MB * task_num (task_num will be 3 if there are three tasks), and by GR is around 70.13MB. The memory required by all these systems are very small as compared to the total memory available in a modern computer.

• Our data generator DG has around 1,460,361 parameters (around 44.57MB, calculated by 32 Byte per parameter), which is fixed. Saving data would need much more memory. Taking the MNIST dataset as an example. It has 60,000 training samples and the size of each sample is 28*28. The memory needed to store the data is 28*28*60,000 = 47,040,000 (around 358.89MB, calculated by 8 Byte per unit). And this number multiplies when the number of tasks increases.

**Training and Test Time**

Taking the CIFAR10 dataset as an example (others are similar), the total time used by our model and baselines are shown in Table 6. We can see that our method needs a bit more time than baselines but not too much (under 25% for training and under 28% for testing compared with GR). This is a small price to pay for the major gain in accuracy and in dealing with the catastrophic forgetting problem.

**Accuracy on the Last Task**

Table 6: Empirical evaluation of different components

| *Time* | *EWC* | *IMM* | *GR* | *Our PGMA* |
|---|---|---|---|---|
| *Training Time/per epoch (s)* | 2.460 | 4.930 | 8.673 | 10.836 |
| *Test time(s)* | 0.703 | 0.707 | 0.728 | 0.930 |

ADAM gets the best accuracy on the last task as Adam optimizer learns the new task only and does not need to worry about the forgetting problem on the old tasks. For the last task, our PGMA system obtained comparable accuracy on average with the baselines designed for overcoming forgetting. However, we are much better at preventing forgetting of old tasks. The results for the last task are given in Table 7. Note that although EWC is quite good at the last task but it suffers seriously from forgetting as shown in Table 1 or Figure 3, which is consistent with results reported in other papers.

Table 7: Accuracy of the last task after all tasks have been learned.

| Model | shuffled MNIST | shuffled MNIST | disjoint MNIST | disjoint MNIST | disjoint CIFAR-10 | THUCNews | DBPedia | DBPedia | DBPedia |
|---|---|---|---|---|---|---|---|---|---|
| | 3 tasks | 5 tasks | 2 tasks | 5 tasks | 2 tasks | 2 tasks | 2 tasks | 3 tasks | 5 tasks |
| Adam | 98.30 | 98.27 | 97.28 | 95.78 | 83.98 | 93.56 | 95.30 | 97.52 | 98.45 |
| EWC | 98.12 | 97.99 | 96.97 | 95.61 | 75.38 | 86.72 | 96.11 | 96.55 | 93.32 |
| GR | 98.10 | 97.18 | 96.34 | 92.31 | 77.82 | 87.20 | 93.18 | 89.40 | 98.76 |
| IMM | 97.06 | 95.36 | 95.07 | 86.94 | 48.74 | 65.08 | 93.84 | 85.72 | 70.19 |
| Our PGMA | 98.15 | 97.26 | 96.22 | 93.47 | 73.75 | 86.36 | 96.01 | 91.14 | 82.99 |

