# OpenReview forum: "Overcoming Catastrophic Forgetting for Continual Learning via Model Adaptation"
_ICLR.cc/2019/Conference_

### Official Review · AnonReviewer1 · 2018-10-28
**Interesting ideas and results. Needs improvement in writing, formalization and possibly more ablation studies.**

**Rating:** 7
**Confidence:** 4

**Review:**

This paper proposes a Dynamic Parameter Generator (DPG) that given a test input modifies the parameters of a classification model. They also propose to regularize the training using a Data Generator (DG) to slow down catastrophic forgetting. DG is used to constrain the training that the internal representations of data generated by DG does not rapidly change. DG removes the need for storage of data or labels.

Positives:
- Both ideas of DPG and DG are novel in preventing catastrophic forgetting.
- DG is novel because it does not require storage of data and does not depend on labels.
- Experimental results are significantly better than the previous state-of-the-art.

Suggestions and clarification requests:
- Figures are very small and equations are cramped because of reduced spacing.
- There are some vague explanations in the intro that could be reduced. It would be nice to first introduce concrete math then give the intuitions. That saves some space.
- It would nice to compare to the recent Progress & compress [1]. Unfortunately, they have not provided results on benchmark MNIST tasks.
- This work is related to a recently proposed idea in architecture search [2] that learns to predict the weights of a network given its architecture.
- Can you clarify whether you have used DG at test time?
- Can you report results without using DG? It is not clear whether DPG is accountable for preventing the catastrophic forgetting or the sluggishness enforced by DG.
- Questions 1 and 2 need more formalization if the authors want to clearly prove a statement.
- As the answer to Question 1 suggests, have you explored enforcing a Lipschitz constraint?
- The answer to Question 2 is interesting. Could you rewrite it more formally? It seems like you can argue that DG’s objective encourages the employment of unused parameters which is important in tackling catastrophic forgetting.
- Can you elaborate on how much forgetting happens for DG?
- It seems that in figure 3.f and 3.c the MA method is unable to reach the best possible performance on the last task. Can you also report the table of accuracies on the last task?

[1] Schwarz, Jonathan, et al. "Progress & Compress: A scalable framework for continual learning." arXiv preprint arXiv:1805.06370 (2018).
[2] Brock, Andrew, et al. "SMASH: one-shot model architecture search through hypernetworks." arXiv preprint arXiv:1708.05344 (2017).

---

> ### Author Response · Authors · 2018-11-10
> **Thank you for your positive and constructive feedback （Part one）**
>
> Thank you for your positive and constructive feedback. We apologise for the slow response as one of the authors is traveling. Below, we respond to each of your comments.
>
> 1.	Re: “small figures” and “There are some vague explanations in the intro that could be reduced”
>
> Thanks for your suggestions. We will enlarge the figures and improve the intro.
>
> 2.	Re: “It would nice to compare to the recent Progress & compress [1]. Unfortunately, they have not provided results on benchmark MNIST tasks.”
>
> Yes, we will consider to use this method as a baseline. As we could not find their code on the Internet, it is hard to compare with it in the response period.
>
> 3.	Re: “Can you clarify whether you have used DG at test time?”
>
> No. DG is only used in training. It is not used at the test time.
>
> 4.	Re: “Can you report results without using DG? It is not clear whether DPG is accountable for preventing the catastrophic forgetting or the sluggishness enforced by DG.”
>
> DG is an essential component for our framework. We cannot remove DG since our method won’t work without it. We showed that DPG is very useful in preventing catastrophic forgetting by the ablation study in Table 3.
>
> 5.	Re: “This work is related to a recently proposed idea in architecture search [2]”
>
> Thanks for pointing out this. We will cite and discuss [2] in the related work. The parameter generations in the two papers are for very different purposes. Paper [2] aims to accelerate architecture selection by learning an auxiliary HyperNet that generates weights for a main model conditioned on that model’s architecture. Our method aims to use the generated parameters for model adaption, which greatly reduces the parameter generation dimensions.
>
> 6.	Re: “Questions 1 and 2 need more formalization if the authors want to clearly prove a statement.” and “The answer to Question 2 is interesting. Could you rewrite it more formally? It seems like you can argue that DG’s objective encourages the employment of unused parameters which is important in tackling catastrophic forgetting.”
>
> Thank you for your suggestion. We will rewrite and add more formalization to make these two questions clearer.
>
> 7.	Re: “As the answer to Question 1 suggests, have you explored enforcing a Lipschitz constraint?”
>
> We have not enforced a Lipschitz constraint in our experiments. We give the proof of Question 1, aiming to provide a theoretical guarantee and an upper bound for using the replayed data x’ in Formula 7. But thanks for the good suggestion and we will explore this experiment.
>
> **Due to the limitation of character number, we divide our responses into two parts, please also see the second part.**

---

> > ### Comment · AnonReviewer1 · 2018-12-08
> > **Thanks for clarifications and rewrite**
> >
> > My questions are answered and thanks for incorporating some in the document.

---

> ### Author Response · Authors · 2018-11-10
> **Thank you for your positive and constructive feedback （Part two）**
>
> 8.	Re：“Can you elaborate on how much forgetting happens for DG?”
>
> We have two understandings for your question. The first one is how much forgetting happens for DG before using constraints. DG suffers from the same forgetting problem as the classifier (solver in our paper) as it is also trained using the same SGD-like method, which only optimizes the current task without considering the previous tasks. The second is how much forgetting happens for DG after using the constraints. This is hard to measure. However, the degree of forgetting is indirectly reflected in the final accuracy. Furthermore, we obtained a better result for using DG + constraints only (without DPG) than GR, which shows that our method suffers less forgetting.
>
> 9.	Re: “It seems that in figure 3.f and 3.c the MA method is unable to reach the best possible performance on the last task. Can you also report the table of accuracies on the last task?”
>
> Yes, ADAM gets the best accuracy on the last task as Adam optimizer learns the new task only and does not need to worry about the forgetting problem on the old tasks. For the last task, Our MA system obtained comparable accuracies on average with the baselines designed for overcoming forgetting. However, we are much better at preventing forgetting of old tasks. The results for the last task are given in the table below. Note that although EWC is quite good at the last task but it suffers seriously from forgetting as shown in Table 2 or Figure 3, which is consistent with results reported in other papers.
> -----------------------------------------------------------------------------------------------------------------------------------------------
>                |                                                  Last task  Accuracy (%)
>                | ---------------------------------------------------------------------------------------------------------------------------------
>   Model | Shuffle MNIST | Disjoint MNIST | Disjointed CIFAR10 | THUCNews | DBPedia77| DBPedia554
> ------------------------------------------------------------------------------------------------------------------------------------------------
>  ADAM  |        98.30           |         97.28          |           83.98                |       93.56      |      95.30     |       97.52
>   EWC    |        98.12           |         96.97          |           75.38                |       86.72      |      96.11     |       96.55
>     GR     |        98.10           |         96.34          |           77.82                |       87.20      |      93.18     |       89.40
>   IMM    |        97.06           |         95.07          |           48.74                |       65.08      |      93.84     |       85.72
> Our MA|        98.15           |         96.22          |           73.75                |       86.36      |      96.01     |       91.14
> ------------------------------------------------------------------------------------------------------------------------------------------------

---

> ### Author Response · Authors · 2018-11-27
> **Thank you**
>
> Thank you. We have revised the paper according to your comments and improved the clarity.  During the revision, we re-organized and re-wrote the paper and also asked a colleague who was not familiar with the work to read and understand. We hope the main idea is now clear. If you have time, please check the revision, thanks again.

---

### Official Review · AnonReviewer3 · 2018-11-02
**Unclear description of algorithm, experiments could be improved.**

**Rating:** 6
**Confidence:** 2

**Review:**

Summary:
- In this paper, an algorithm to improve the catastrophic forgetting of the model is proposed. The key idea consists of 1) introducing the dynamic parameter generator (DPG) for "model's adaptation" to data at test time and 2) data generator (DG) for remembering previously trained dataset.

Pros:
- Empirical results seem strong. The proposed result outperforms existing algorithms by quite large margin.

Cons:
- In general, I felt that the paper is unorganized and hard to read. Clarity should be definitely improved if this paper is to be published as a conference paper.

- Output of dynamic parameter generator is very high dimensional (it requires weight with dimension of input dim x NN weight dim). I think this approach is not scalable to higher dimension and typically requires even more memory than storing the whole dataset.

- Although auto-encoder and generative replay was considered to reduce memory consumption, there is no description of how much memory is saved by them. In order to make the argument more convincing, the authors should explicitly describe the amount of memory consumed by each algorithms.

- There seems to be a lot of ideas introduced, i.e, DPG for generation of weights, auto-encoders for generation of data and layer output constraint, i.e., Equation (7).  I think each of introduced method deserves some amount of empirical evaluation to validate its contribution to the performance.

- Experiments only consider 2~3 tasks, which does not seem very representative for the lifelong learning tasks.

---

> ### Author Response · Authors · 2018-11-10
> **Thank you for your comments.**
>
> Thank you for your comments. We apologize for the slow response as one of the authors is traveling.
> We are happy to discuss to clear some confusions. Below, we respond to each of your comments. We appreciate your further feedback.
>
> 1.	Re: "the paper is unorganized and hard to read”
>
> We will improve the organization of the paper to make everything clearer.
>
> 2. Re: "Output of dynamic parameter generator is very high dimensional"
>
> If we generate all the parameters in the model, the output will be very high dimensional. However, there is no need to generate all the parameters. We show that it suffices to generate and replace only the parameters in the last layer, which is very low dimensional (e.g., in our MNIST experiment setting, the dimension is around 200*60). We discuss this operation in equation 4 and its context. Table 3 also shows that our approach can achieve very good results by only replacing a portion of the parameters of the last layer.
>
> 3. Re: "description of how much memory is saved"
>
> Saving data would need much more memory. Taking the MNIST dataset as an example. It has 60,000 training samples and the size of each sample is 28*28. The memory needed to store the data is 28*28*60,000 = 47,040,000. This number multiplies when the number of tasks increases, while our DG has only around 1,460,361 parameters, which is fixed.
>
> 4. Re: "each of introduced method deserves some amount of empirical evaluation"
>
> Thank you for pointing out that. The third column in Table 3 (our model replacing 0% parameters) can be regarded as removing the DPG in our model. We can see a significant drop compared with the seventh column (replacing 80% parameters), which shows the effectiveness of DPG. The third column in Table 3 shows the performance of our model when only DG and constraints are used. We have conducted a new experiment for our model without using constraints (DG+label(like GR)) but replacing them with the replay method (which needs to predict the labels as we discussed in the third paragraph to the last in Section 1). We get the accuracy of 0.8821 ± 0.0281, which is poorer than DG+Constraint (90.96±0.69) and DPG+DG+Constraints (96.07±0.62). For DG, we can’t remove it because our method won’t work without it.
>
> 5. Re: "Experiments only consider 2~3 tasks".
>
> We used the same setting in terms of the number of tasks as the previous works (EWC and IMM) for easy comparison. We have just conducted additional experiments on 5 tasks. Here are the results. (1) On shuffled MNIST, the accuracy of GR is 94.536, the accuracy of IMM is 96.088, and the accuracy of our MA is 96.774 (GR and IMM are the best baselines, see Table 2). (2) On disjoint MNIST (2 classes per task), the accuracy of GR is 75.473, the accuracy of IMM is 67.250, and the accuracy of our MA is 81.700. (3) On disjoint DBPedia (3,3,3,3,2 classes in the 5 tasks), the accuracy of GR is 63.714, the accuracy of IMM is 64.040, and the accuracy of our MA is 69.680. As you can see, our method is still markedly better than the baselines.

---

> ### Author Response · Authors · 2018-11-27
> **Thank you**
>
> Thank you. We have revised the paper according to your comments and improved the organization.  If you have time, please check the revision, thanks again.

---

### Official Review · AnonReviewer2 · 2018-11-13
**paper needs major revision to improve clarity and empirical validation**

**Rating:** 5
**Confidence:** 4

**Review:**

This paper proposes a method for continual learning. The model has three components: a) a data generator to be used at training time to replay past examples, b) a parameter generator that takes the input observation to produce parameters for c) the actual classifier. The authors demonstrate the method on simple datasets with a stream of 2 or 3 tasks.

Strenghts:
- the combination of components is novel
- the method does not rely on task descriptors neither at training nor test time
Weaknesses:
- the paper needs a major rewrite to improve fluency and to better organize and describe the proposed approach
- the empirical validation is weak.

Relevance
Learning in a continual setting is certainly very relevant for this venue.

Novelty
While each component is by itself not very novel (replay methods for continual learning have already been used, networks predicting parameters have also become a fairly common approach in meta-learning literature), the proposed combination is novel in this sub-field.

Clarity
Clarity is very poor and definitely does not meet the acceptance bar for this conference. I believe that the authors would need to make a major revision to address this issue. While ICLR allows authors to revise papers, I think the revision needed to fix this draft goes beyond the acceptable limit, as reviewers would then need to make a whole new revision.
First, fluency is very poor. There are lots of grammatical errors (see first sentence of introduction "neural networks suffers.."),  a plethora of un-necessary acronyms which force the reader to go back and forth to figure out what they refer to (MA, DG, DPG, DPG&S, ...), and several sentences are not well formed (e.g., read first sentence of introduction).
Second, some statements are contradictory; e.g., the authors define "basic unit" as "simple MLP with one hidden layer", but then say it "is an activation function plus a matrix transformation"..
Third, graphics and formulas are too small and not legible.
Fourth, the organization of the paper is poor, it is very wordy yet vague. For instance, the authors should precisely describe how the data generator is trained in sec. 2.3. The authors should provide an algorithm summarizing how the different components interplay both at training and test time. At present, I am making educated guesses about how this system works.
For instance, how are real and generated examples interleaved? how is forgetting prevented in the data generator?

References to prior work
While there are lots of references in sec. 4, they are not sufficiently well described - see third paragraph of sec. 4 where the authors cite almost 20 papers by simply saying they are "some other approaches".
Also, I did not find mention to methods predicting parameters in the meta-learning community but also others like:
Denil et al. "Predicting network parameters in deep learning" NIPS 2013

Empirical validation
The empirical evaluation does show an advantage of the proposed approach on some simple streams composed by up to three tasks. However, a) the tasks are really simple because of the small number of tasks considered and b) the baselines are weak. For instance, EWC is now a bit out-dated as there are variants that work a bit better, like:
Chaundry et al. "Riemannian Walk ..." ECCV 2018
and there are other methods like "Progress and Compress" Schwartz ICML 2018 the authors could have compared against.
Besides, the authors do not mention anything about memory and time cost both at training and test time, possibly including the time to cross validate all the hyper-parameters of this method.
Overall, I am left with the sense that the proposed approach will be hard to scale to many more tasks and more realistic images (for which we do not quite know how to train efficiently and well generators).

Other comments
I did not find the formalization in eq. 9 very useful. The first and last term in that equation can be very big and there is no sense of how lose this bound is.
Also, it is not clear whether there is a general principle to partition the set of parameters (to determine which ones should be shared).

---

> ### Author Response · Authors · 2018-11-16
> **Thank you for your comments.**
>
> Thank you for your comments. We are happy to discuss to clear some confusions. Below, we respond to each of your comments. We appreciate your further feedback.
>
> 1.	Re: “While ICLR allows authors to revise papers, I think the revision needed to fix this draft goes beyond the acceptable limit, as reviewers would then need to make a whole new revision.”
>
> That is sad. It seems you will ignore our response and won’t reconsider your decision. The clarity issue that you mentioned is not difficult to address. We will improve the organization of the paper to make everything clearer.
>
> 2.	Re: “the tasks are really simple because of the small number of tasks considered”
>
> We used the same setting in terms of the number of tasks as the previous works (EWC and IMM) for easy comparison. We have just conducted additional experiments on 5 tasks. Here are the results.
> (1) On shuffled MNIST, the accuracy of GR is 94.536, the accuracy of IMM is 96.088, and the accuracy of our MA is 96.774 (GR and IMM are the best baselines, see Table 2).
> (2) On disjoint MNIST (2 classes per task), the accuracy of GR is 75.473, the accuracy of IMM is 67.250, and the accuracy of our MA is 81.700.
> (3) On disjoint DBPedia (3,3,3,3,2 classes in the 5 tasks), the accuracy of GR is 63.714, the accuracy of IMM is 64.040, and the accuracy of our MA is 69.680. As you can see, our method is still markedly better than the baselines.
>
> 3.	Re: “the baselines are weak. For instance, EWC is now a bit out-dated as there are variants that work a bit better”
>
> EWC is indeed an early system, but it is used as a baseline by most papers on continual learning including "Progress and Compress" Schwartz ICML 2018. Please notice that we have not only compared with EWC. IMM and GR are not weak or out-dated baselines.  Also, we will compare those papers you recommended, thanks.
>
> 4.	Re: “the authors do not mention anything about memory and time cost both at training and test time, possibly including the time to cross validate all the hyper-parameters of this method.”
>
> Memory
> 1)	The memory used by our model is around 151.49MB, by IMM is around 38.96MB * task_num (task_num will be 3 if there are three tasks), and by GR is around 70.13MB. The memory required by all these systems are very small as compared to the total memory available in a modern computer.
> 2)	Furthermore, our data generator DG has around 1,460,361 parameters, which is fixed. Saving data would need much more memory. Taking the MNIST dataset as an example. It has 60,000 training samples and the size of each sample is 28*28. The memory needed to store the data is 28*28*60,000 = 47,040,000. And this number multiplies when the number of tasks increases.
>
> Training and Test time
> Taking the CIFAR10 dataset as an instance (others are similar), the total time used by our model and baselines are shown below. We can see that our method needs a bit more time than baselines but not too much (under 25% for training and under 28% for testing compared with GR)
>
>                                         Time      |      IMM     |      GR     |      Ours
> ----------------------------------------------------------------------------------------------
> Training Time/per epoch (s)     |      4.93      |    8.673  |    10.836
> ----------------------------------------------------------------------------------------------
>                              Test time(s）   |     0.707     |    0.728  |     0.930
>
> We did not give the memory and time costs in the paper as existing papers on the topic generally do not give such numbers.
>
> 5.	Re: “it is not clear whether there is a general principle to partition the set of parameters (to determine which ones should be shared).”
>
> Based to our ablation experiments (see Table 3), replacing any portion of the parameters (20%, 40% …) in our model outperforms the baselines. Cross validation can be used to find the best partition.

---

> > ### Comment · AnonReviewer2 · 2018-11-27
> > **thank you**
> >
> > Thank you for your response and additional results.
> > I still think that lack of clarity is the major issue of the current draft.

---

> > > ### Author Response · Authors · 2018-11-27
> > > **Thank you**
> > >
> > > Thank you. We have revised the paper according to your comments and improved the clarity.  During the revision, we re-organized and re-wrote the paper and also asked a colleague who was not familiar with the work to read and understand. We hope the main idea is now clear.  If you have time, please check the revision, thanks again.

---

> > > > ### Comment · AnonReviewer2 · 2018-12-06
> > > > **read the revision: raising my score but not recommending acceptance**
> > > >
> > > > Dear Authors,
> > > >  I have read the revision. The draft is more clear now. I am not strongly opposed to accept this work anymore, although I am still hesitant because I feel that more rewrite is necessary before publication.
> > > >
> > > > The organization improved but it is still not good enough, there is a lot of back and forth between the description of the different parts of the model and how they are trained. It would have been better to perhaps formalize more, and explain what input and outputs of each components are, what the loss function is and how things are trained all together. At present, things are still scattered and I would honestly still do not know how to reproduce this work just based on your model description.
> > > > Still there are lots of references in the related work but little relation to your own work. Sentences like "Our method is very different from these existing approaches." are not sufficient to relate this work to prior work!
> > > >
> > > > More fundamentally, this approach has interesting aspects because it does not require the specification of the task. Yet it feels premature, given how many parts are needed to make it work and the clumsiness of the training procedure. Given how many components there are and given how they are all prone to forget on their own, there are a lot of dependencies which are hard to control and can make training not robust.
> > > >
> > > > The empirical results show this method works better than the models considered. Still the setting is rather toy-ish and the gains are often very small.

---

> > > > > ### Author Response · Authors · 2018-12-08
> > > > > **Thank you**
> > > > >
> > > > > Thank you for taking the time to read our revision and giving the new comments and suggestions. We will improve the related work and organization more in the next version. With your suggestions, we will make the paper very clear. We are also ready to release the code and hope that can ease your concern about “reproducibility”.
> > > > >
> > > > > We feel that our approach is not difficult to train, and the confidence interval given in Tables 3 and 4 also shows the robustness of our approach. Although our framework involves three components (DG, DPG and solver S), for training there are only two major components (DG and DPG&S) as DPG and S are trained together. Each major component has its own independent objective and is not intractable. Data Generator (DG) is adapted from WAE [1], which is a theoretically guaranteed method and performs robustly according to many prior works, e.g., [1] and [2]. Several existing methods have a few components too, i.e., GR and IMM (our two baselines), widely used GAN [3], and multi-task learning and multi-objective learning systems [4].
> > > > >
> > > > > Re: “Still the setting is rather toy-ish and the gains are often very small.”
> > > > >
> > > > > The two image datasets are quite widely used benchmark datsets (we have 2 classes per task, which is already the limit). To further show generality, we used two text datasets that are not used by existing work. We would really like to draw your attention to the trend. As our system consistently outperforms the baselines for 2, 3, 5 tasks and with more tasks our method improves more, it is reasonable to believe that this trend will continue with more tasks. So we believe that our setting should not be considered “toy-ish.” We also believe that the gains are not “very small.” For example, from the results shown in Table 2, we improve the score (1) from 75.47% to 81.70% for disjoint MNIST (5 tasks); (2) from  64.04% to 69.68% for DBPedia (5 tasks); (3) from 32.36% to 40.47% for disjoint CIFAR-10 (5 tasks); (4) from 47.35% to 52.93% for THUCNews (5 tasks). Experimental results shown in Table 1 have similar trends too. We hope that you can reconsider again. Thank you.
> > > > >
> > > > >
> > > > > [1] Tolstikhin, I., Bousquet, O., Gelly, S. and Schoelkopf, B., 2017. Wasserstein auto-encoders. ICLR, 2018
> > > > >
> > > > > [2] Bahuleyan, H., Mou, L., Vamaraju, K., Zhou, H. and Vechtomova, O., 2018. Probabilistic natural language generation with wasserstein autoencoders. arXiv preprint arXiv:1806.08462.
> > > > >
> > > > > [3] Goodfellow, I., Pouget-Abadie, J., Mirza, M., Xu, B., Warde-Farley, D., Ozair, S., Courville, A. and Bengio, Y., 2014. Generative adversarial nets. In Advances in neural information processing systems (pp. 2672-2680).
> > > > >
> > > > > [4] Ozan Sener, Vladlen Koltun. Multi-Task Learning as Multi-Objective Optimization. NeuralPS, 2018.

---

### Public Comment · (anonymous) · 2018-11-07
**why dynamic parameter generator works ?**

Comment: in section 2.4 question2, authors said " If X0 k is a row low rank matrix, w∗i,k can be trained to ﬁt the new tasks. Otherwise, especially if X0 k is a row full rank matrix, w∗i,k cannot be trained and therefore cannot learn new tasks. However, beneﬁting from dynamic parameter generation, our approach will not suffer from this problem. "

 I could not understand this well. In my understanding  w∗i,k is generated from DPG parameters, when the X matrix is full rank, the author said that  w∗i,k can not be trained , why the parameters which used to generate w∗i,k  can be trained ? Thanks.

---

> ### Author Response · Authors · 2018-11-07
> **Thanks for your interest in our paper and a good question.**
>
> It is the adaptability of our model on different tasks that avoids the issue you quoted. Please see formula 8 and the explanation for question 2 in the paper. The stacked X denotes the input of the layer where the constraints are used,
> and w∗i,k cannot be changed if X is a row full rank matrix; otherwise formula 8 will get a huge loss and the w∗i,k learned from old tasks cannot be trained to fit the new samples without forgetting. However, in our model, a part of w∗i,k is generated by DPG, which means that each sample from a different task has a different w∗i,k. In that case, we can train DPG such that without w∗i,k for samples from old tasks being changed, the new task can still be learned because its samples get their own w∗i,k’s. In other words, constraints on w∗i,k for samples from old tasks won't interfere with the training of w∗i,k in the new task.
>
> If you still have any doubts, please feel free to tell us.
> Thanks again.

---

### Meta-Review · Area_Chair1 · 2018-12-15

**Confidence:** 3
**Recommendation:** Accept (Poster)

**Metareview:**

This paper presents a promising model to avoid catastrophic forgetting in continual learning. The model consists of a) a data generator to be used at training time to replay past examples (and removes the need for storage of data or labels), b) a dynamic parameter generator that given a test input produces the parameters of a classification model, and c) a solver (the actual classifier). The advantages of such combination is that no parameter increase or network expansion is needed to learn a new task, and no previous data needs to be stored for memory replay.

There is reviewer disagreement on this paper. AC can confirm that all three reviewers have read the author responses and have significantly contributed to the revision of the manuscript.

All three reviewers and AC note the following potential weaknesses: (1) presentation clarity needed substantial improvement. Notably, the authors revised the paper several times while incorporating the reviewers suggestions regarding presentation clarity. R2 has raised the final rating from 4 to 5 while retaining doubts about clarity.
(2) weak empirical evidence: evaluation with more than three tasks and using more recent/stronger baseline methods would substantially strengthen the evaluation (R2, R3). AC would like to report the authors added an experiment with five tasks and provided a verbal comparison with "Riemannian Walk for Incremental Learning: Understanding Forgetting and Intransigence", ECCV-2018 by reporting the authors results on the MNIST dataset.
(3) as noted by R2, an ablation study of different model components could strengthen the evaluation. The authors included such ablation study in Table 4 of the revised paper.
(4) reproducibility of the model could be difficult (R1). In their response, the authors promised to make the code publicly available.

AC can confirm that all three reviewers have contributed to the final discussion. Given the effort of the reviewers and authors in revising this work and its potential novelty, the AC decided that the paper could be accepted, but the authors are strongly urged to further improve presentation clarity in the final revision if possible.